# Photosynthetic Microorganisms and Biogenic Synthesis of Nanomaterials for Sustainable Agriculture

**DOI:** 10.3390/nano15130990

**Published:** 2025-06-26

**Authors:** Svetlana Codreanu, Liliana Cepoi, Ludmila Rudi, Tatiana Chiriac

**Affiliations:** Institute of Microbiology and Biotechnology, Technical University of Moldova, MD-2004 Chisinau, Moldova; liliana.cepoi@imb.utm.md (L.C.); ludmila.rudi@imb.utm.md (L.R.); tatiana.chiriac@imb.utm.md (T.C.)

**Keywords:** cyanobacteria, microalgae, bionanotechnology, nanosynthesis, nanoparticles, agriculture

## Abstract

Sustainable agriculture faces increasing challenges, necessitating innovative approaches to advance resource efficiency with minimal ecological consequences. One promising solution is nanobiotechnology, which takes advantage of natural systems for the eco-friendly synthesis of functional nanomaterials. Prokaryotic cyanobacteria and eukaryotic microalgae, due to their rapid growth, adaptability to diverse environments, and capacity for biosynthesis of valuable compounds, are model organisms highly suitable for medical, biotechnological, industrial, agricultural, and environmental applications. These photosynthetic microorganisms have demonstrated their efficacy in the biosynthesis of nanomaterials, which has potential benefits in various agricultural applications. The use of cyanobacteria- and microalgae-based nanomaterials in improving agricultural practices represents an emerging field of nanotechnology that requires ongoing research and responsible application management. To present a complete and timely foundation for this field, a systematic review of relevant research from the last five years was performed, exploring the contribution of cyanobacteria and microalgae to the advancement of nanobiotechnology as an efficient biotransformative tool for sustainable agriculture.

## 1. Introduction

Nanotechnology, a cutting-edge field that utilizes nanoscale materials (nm) to develop valuable products with diverse applications, is increasingly recognized as a powerful driver of innovation across sectors such as healthcare, electronics, catalysis, and the agri-food industry. Interest in nanomaterials (NMs), particularly nanoparticles (NPs), has grown rapidly in recent years due to their unique physicochemical properties compared to bulk materials [1]. To address the challenges of sustainable agriculture—balancing increased food demand with environmental preservation and long-term resilience—researchers increasingly employ bio-based synthesis of nanoparticles. Biosynthesis using the inherent reducing capabilities of biological compounds in living organisms offers a viable and environmentally friendly alternative for nanoparticle production [2]. Photosynthetic microorganisms offer significant potential for bio-nanosynthesis due to their broad environmental adaptability, particularly their capacity to tolerate and accumulate heavy metals [3]. Throughout evolution, cyanobacteria and microalgae have developed various biochemical and molecular protective mechanisms, including metal elimination, extracellular and intracellular sequestration, biosorption, precipitation, production of chelating agents, bioaccumulation, biotransformation by specific enzymes and metabolites, antioxidant defense systems, and regulation of metal transport and detoxification genes [4,5]. Moreover, their phototrophic nature, rapid growth, and high photosynthetic efficiency in producing diverse bioactive compounds make them a promising source of biostimulants, biofertilizers, and biopesticides [6]. The capacity of cyanobacteria and microalgae to accumulate metals and mediate their transformation into nanoparticles establishes them as highly effective bionanofactories for eco-friendly synthesis [7]. Researchers are exploring the use of these microorganisms to produce various nanomaterials—including metal nanoparticles, metal oxide nanoparticles, and nanocomposites—through environmentally friendly and sustainable processes. These biogenic nanomaterials exhibit diverse physicochemical and biological properties, supporting applications ranging from improving soil health and enhancing crop growth to controlling pests and pathogens [3]. The synthesis of nanomaterials for agricultural applications using cyanobacteria and microalgae represents an emerging area of research that integrates the biological capabilities of these microorganisms with advances in nanotechnology. The potential of these organisms for bio-nanosynthesis remains largely unexplored [8]. Furthermore, while well established in medicine, the use of nanomaterials in agriculture, particularly for plant growth and disease control, is still in its early stages and demands further investigation [9].

Given the rapid evolution of bionanotechnology and its growing importance in agriculture, this systematic review intends to offer a comprehensive summary of recent studies (from the past five years) on the green synthesis of nanomaterials using cyanobacteria and microalgae, with a particular focus on their agricultural applications.

## 2. Materials and Methods

This systematic review was carried out following “The PRISMA 2020 Statement: An Updated Guideline for Reporting Systematic Reviews” [10]. The review is grounded in research published across the following databases: Springer Link, ScienceDirect, MDPI, Frontiers, Research4Life, Taylor & Francis, and Oxford Academic, addressing the application of cyanobacteria—and microalgae-based nanosynthesis in agriculture. The identification of relevant studies was supplemented by manual citation searching.

Search terms and languages. The list of relevant search terms for targeted articles included “cyanobacteria”, “microalgae”, “nanoparticle synthesis”, and “agriculture”. The first two terms were combined with others using the Boolean operators “OR” and “AND”. No language limitations were imposed on the electronic search. Still, the manual search was limited to English, French, and Romanian publications.

Articles screening. The screening of publications was conducted in two distinct stages, applying predefined eligibility criteria: (1) title and abstract screening and (2) full-text screening.

Eligible Topics. Eligibility included any publication that addressed at least two of the following topics: “Biotechnology of cyanobacteria and microalgae”, “Bio-nanosynthesis”, and “Applications in agriculture”. The application of these eligibility criteria enabled the classification of publications into four groups, as illustrated by a Venn diagram (Figure 1a).

Eligible Organisms. The inclusion of photosynthetic microorganisms in this study reflects a compromise between the conventional and phylogenetic classifications of cyanobacteria and microalgae. The organisms considered are microscopic prokaryotic and eukaryotic primary producers that utilize sunlight to convert inorganic substances into organic compounds.

In this context, we also acknowledge that, in some publications, “Microalgae” refers to both eukaryotic microalgae and prokaryotic cyanobacteria (often referred to as blue-green algae). In other publications, microalgae are classified under “algae”, while cyanobacteria are listed alongside other bacterial species. Purple bacteria are not included in this study.

## 3. Results

A comprehensive electronic literature search across seven databases, supplemented by hand searching, using the strategy outlined in the PRISMA flow diagram (Figure 2), identified 329 relevant articles. A total of 178 articles fulfilled the inclusion criteria and were selected for the systematic review.

Continuing with the grouping of eligible study topics, the selected publications were categorized into four thematic groups (Figure 1b), which formed the basis for the descriptive analysis presented below.

An analysis of the selected publications revealed that the most prevalent are scientific reviews, primarily addressing the general properties of nanoparticles, characterization methods, and biosynthesis mechanisms. Original research articles constitute only 24.5% of the total. The time distribution of these publications is relatively balanced, with a noticeable trend toward a more even distribution between publication types in 2024 (Figure 3).

Our review primarily focuses on publications from the past five years related to photosynthetic microorganisms and their contribution to advancements in nanobiotechnology for agricultural applications.

### 3.1. Cyanobacteria and Microalgae for Agriculture

Photosynthetic microorganisms, one of the most important groups of microorganisms on the planet, represent a valuable resource for sustainable ecosystems and a promising tool for sustainable agriculture.

The importance of cyanobacteria and microalgae in sustainable agriculture consists of the synergetic effect of these microorganisms with the following: (1) plant metabolism, improving crop productivity through the N_2_ fixation and production of secondary metabolites; (2) soil fertility, enhancing soil water holding capacity, solubilization, and mobility of nutrients, increasing organic matter content and removing pollutants; and (3) atmosphere health, reducing greenhouse gases by CO_2_ sequestration, assimilation in biomass and conversion to biofuel [11].

Biofertilizers, stimulants, and pesticides derived from cyanobacteria and microalgae have the potential to become essential elements of sustainable agriculture, contributing to solutions for global challenges such as food security, climate change, and environmental degradation [12,13]. Figure 4 summarizes the mode of action and impact of these bio-based products.

The use of photosynthetic microorganisms to improve agricultural practices is widely discussed in the scientific literature. Prokaryotic cyanobacteria and eukaryotic microalgae share similar habitats, morphology, and the ability to perform oxygenic photosynthesis. The pigments in both groups are part of a diverse array of metabolites that contribute to their adaptability across various environments. Their presence in different soil types is well documented and often yields beneficial effects in various ecological contexts [14,15]. These remarkable organisms offer valuable benefits to modern agriculture, including enriching soil nutrients, enhancing the uptake of macro- and micronutrients, and improving soil texture, structure, and water retention capacity. Many of these benefits are associated with diazotrophs—a group of nitrogen-fixing cyanobacteria with specialized cells (heterocysts) that serve as nitrogen storage sites and can be used as eco-friendly biofertilizers. Diazotrophic cyanobacteria can contribute approximately 20–30 kg N ha^−1^ annually through nitrogen fixation and improve phosphorus bioavailability by solubilizing and mobilizing insoluble organic phosphates. Major nitrogen-fixing organisms include *Anabaena* sp., *Aulosira fertilissima*, *Scytonema* sp., *Nostoc linckia*, *Tolypothrix* sp., *Chroococcus* sp., *Calothrix* sp., etc. [5,14,16,17]. In particular, *Nostoc* species are notable for their ability to fix atmospheric nitrogen while producing phytohormones, primarily cytokinins and auxins [18].

Microalgal and cyanobacterial biomass and their extracts are increasingly valued for their rich nutrient and bioactive compound content, which can significantly boost crop yields. In addition to naturally fertilizing and balancing mineral nutrition, cyanobacteria, and microalgae secrete biologically active molecules, such as osmolytes, phenolic compounds, proteins, vitamins, carbohydrates, fatty acids, amino acids, polysaccharides, carotenoids, and phytohormones, which may act synergistically to enhance plant growth [12,14].

Cyanobacteria and microalgae release metabolites into the culture media in response to both biotic and abiotic stress. To enhance stress tolerance, photosynthetic microorganisms augment the production and excretion of exopolysaccharides, vitamins, antioxidant compounds, etc. For example, high salinity can lead to the production of compatible solutes, such as glycine, betaine, and proline, by cyanobacteria and microalgae [15].

Two photosynthetic microorganisms represent the most promising aquafarming bioeconomy systems. *Arthrospira* (spirulina) and *Chlorella* stand out, accounting for over 90% of global microalgal biomass and drawing growing interest in agriculture [19].

The application of *Arthrospira platensis* bio compounds, obtained by superficial carbon dioxide extraction, confirmed their positive impact on early wheat development and the yield of both wheat and rapeseed, as well as their efficacy against six of nine fungal pathogens tested [20].

Although polysaccharides are key in enhancing soil quality, there is increasing interest in using microalgal extracts or polysaccharides as plant biostimulants. Recent studies have reported their stimulatory effects on tomatoes, petunia, and sugar beet seedlings. Polysaccharides have been shown to activate plant defense pathways in stressed plants, including changes in membrane lipid composition and sterol levels, as well as the activation of defense-related enzymes such as β-1,3-glucanase, phenylalanine ammonia-lyase (PAL), and lipoxygenase [21]. A significant stimulating effect of *Chlorella* extract on the growth of *Medicago truncatula* plants was reported and attributed to phytohormones and algal exopolysaccharides [22]. Likewise, *Chlorella vulgaris* extract enhanced lettuce seedling growth by increasing fresh and dry biomass, chlorophylls, carotenoids, protein content, and shoot ash. Biochemically, it positively affected primary and secondary shoot metabolism, especially nitrogen metabolism [23]. Liquid extracts of *Chlorella vulgaris* and *Spirulina platensis* showed a stimulating effect on the growth of green gram (*Vigna radiata* (L.)) [24]. The investigation of the effects of foliar treatment with *Chlorella vulgaris*, *Arthrospira platensis*, and *Tetradesmus dimorphus* demonstrated the effectiveness of *Chlorella vulgaris* as a biostimulant for enhancing the growth and yield of *Phaseolus vulgaris* [25]. Improved root system development was observed in plants grown under stress conditions, such as nitrogen deficiency when treated with *Chlorella sorokiniana* biomass. Thus, microalgae cultivation for biostimulant production can be applied as a bio-based strategy that effectively improves plant resilience under stress conditions [26].

The current scientific reports highlighted the potential use of algae- and cyanobacteria-based biostimulants and their bioactive compounds in controlling plant-parasitic nematodes, fungi, and oomycetes [27,28]. Regarding their role as biopesticides, microalgae have demonstrated effectiveness as elicitors, activating plant defense responses in roots and leaves (Figure 5). Studies have shown that microalgae induction of defense mechanisms in crops is related to their application in biomass, polysaccharides, exopolysaccharides, lactic acid, or glucosamine [29].

Cyanobacteria and microalgae can become more competitive as biofertilizers and biostimulants by harnessing their bioremediation potential and cultivating them on liquid or solid waste streams.

The authors recommend using these media to produce microalgal cells and specific metabolites, combined with the bioremediation process, as the optimal and affordable solution for waste management and sustainable development [30,31]. A study investigating nutrient removal from landfill leachate using *Chlorella* sp. demonstrated increased algal biomass and removal rates of ammonium, phosphate, and nitrate at 98.7%, 92.7%, and 56.9%, respectively. The resulting algal biomass positively and significantly improved soil quality and can be utilized as an agricultural fertilizer [32]. Biomass ash of *Chlorella vulgaris*, *Chlorella protothecoides*, and *Tetradesmus obliquus* was used for bioremediation of poultry effluents. The produced biomass from *Chlorella vulgaris* was tested as a biostimulant, resulting in a 147% increase in wheat germination index [33]. Another recent study has explored the biostimulant potential of microalgal extracts combined with wastewater treatment. Two cyanobacteria strains (*Synechocystis* sp. and *Phormidium* sp.) and a chlorophyte (*Scenedesmus* sp.) cultured in secondary urban wastewater showed similar phytohormone profiles, with auxin as the most abundant. *Scenedesmus* sp. exhibited the highest nutrient removal rates and phytohormone production, making it the most promising strain for biostimulant production [34]. According to Mahapatra et al., algae derived from wastewater show promise as a complete nutrient source for agriculture and hold potential for use in soilless cultivation systems [35]. Microalgae biomass from wastewater generated by high-tech agricultural systems, such as hydroponics, could be utilized for biofertilizer applications in a closed-loop model, contributing to a circular economy [6].

In their review on photosynthetic microorganisms, biostimulant effects, and prospects for space agriculture, Renaud and Wattiez highlight cyanobacteria and microalgae as producers of secondary metabolites that stimulate plant growth and improve stress tolerance in closed environments, such as those envisioned for space colonization. Additionally, their capacity to produce oxygen and recycle waste further enhances the potential of photosynthetic microorganisms to support life in space [15].

Considering the enormous biotechnological potential of photosynthetic organisms, we believe that bionanotechnology could contribute to their innovative valorization.

### 3.2. Cyanobacteria and Microalgae Applications in Nanotechnology

The most central themes of the analyzed publications include nanosynthesis methods, nanomaterials (NMs) characterization, and the description of their properties.

Joudeh and Linke (2022) [1] provide a comprehensive review of nanomaterials’ classification, physicochemical properties, characterization, and applications. Nanomaterials—the key element of nanotechnology—are classified based on their dimensionality into four categories: (1) zero-dimensional (e.g., nanoparticles, fullerenes), (2) one-dimensional (e.g., nanotubes, nanohorns), (3) two-dimensional (e.g., nanosheets, nanolayers), and (4) three-dimensional (e.g., nanowire arrays, nanotube arrays). Based on their composition, nanomaterials are further divided into (1) organic nanoparticles (e.g., dendrimers, liposomes, micelles, ferritin), (2) carbon-based nanoparticles (e.g., fullerenes, quantum dots), and (3) inorganic nanoparticles, which include metallic nanoparticles, semiconductor nanoparticles, and ceramic nanoparticles (carbonates, carbides, phosphates, and oxides of metals and metalloids). Nanoparticles display mechanical, thermal, magnetic, electronic, optical, and catalytic properties. The cited review summarizes detailed characterization methods and analytical techniques [1]. Compared to bulk materials, nanomaterials exhibit distinct properties, such as increased surface area, which generally enhances their reactivity and expands their range of practical applications.

Nanotechnology applications in many sectors require eco-friendly and energy-efficient methods of nanosynthesis. However, traditional methods of nanomaterial synthesis tend to involve toxic reagents and resource-intensive procedures, posing ecological and health risks. Please refer to Figure 6 for a comparative description of the methods employed in nanosynthesis.

In search of efficient methods for the sustainable synthesis of nanomaterials, researchers have turned to biological systems for inspiration. As noted by Xu L. and colleagues, bioinspired synthesis offers several advantages over chemical synthesis: (a) it is a facile, one-pot process, as bioactive substances act as both reducing and capping agents; (b) it is cost-effective and scalable, relying on inexpensive raw materials and simple procedures; (c) it enables nanomaterial functionalization, improving stability and effectiveness for various applications; and (d) it eliminates the need for toxic or hazardous chemicals, enhancing the biocompatibility of the final product [8].

Bio-based nanosynthesis employs cells and their metabolites to engineer nanoscale structures with notable biocompatibility, sustainability, and environmentally friendly characteristics. Thanks to their ability to accumulate metals and reduce metal ions, photosynthetic microorganisms are ideal candidates for nanomaterial biosynthesis. These organisms are often referred to as “bio-nanofactories” because both their living and dried forms can be used to synthesize nanoparticles [7,36]. Their importance is accentuated by the ability to produce diverse secondary metabolites that help stabilize, cap, and reduce metal precursors to form NPs. As a promising source of biocatalysts, microalgae serve as an effective platform for the biogenic synthesis of both intracellular and extracellular nanoparticles [37,38,39]. The unique set of cyanobacteria and microalgae biocompounds/biomolecules and molecular mechanisms explains the capacity of these organisms to survive in extreme environments and makes them an efficient agent for the green synthesis of NMs [5,40]. A significant number of scientific publications from the last five years review the potential of cyanobacteria and microalgae species as green alternatives for producing NPs, with the major topics discussed presented in Table 1.

As previously mentioned, the systematic position of cyanobacteria described in the reviewed literature is often inconsistent. Despite being prokaryotic organisms, they are frequently referred to as “blue-green algae” and grouped with eukaryotic microalgae in many publications concerning bio-nanosynthesis. Among the publications selected for this review, some focus strictly on cyanobacteria as factories for the green synthesis of NMs [5,33,41,42,43,44]. These review papers analyze possible protocols for nanosynthesis, including concrete examples, a list of cyanobacterial species involved, the synthesis conditions, and characterizations of the obtained NPs.

Most publications analyzed in our review discuss the methods of metallic nanoparticle synthesis by various microalgae species, including cyanobacteria.

Truly interdisciplinary, the authors attribute the use of nanotechnology to harness photosynthetic microorganisms to the emerging fields of science, namely “nano microbiology” and “phyconanotechnology” [45,46,47].

Microalgae are becoming a valuable alternative for nanoparticle synthesis, offering advantages such as rapid growth, low cost, and easy harvesting. Compared to other biological methods, they enable faster nanoparticle production [48]. Additionally, algal filtrates or extracts contain various reducing, capping, and stabilizing agents that can convert metal ions into nanoforms [49]. Therefore, microalgae have become key organisms for the green synthesis of various nanoparticles.

Photosynthetic microorganisms have a remarkable ability to convert metals into nanoforms and can be considered powerful nanofactories for the synthesis of various metallic nanoparticles, such as silver, gold, cadmium, iron, palladium, and platinum, with silver (AgNPs) and gold nanoparticles (AuNPs) as the most prominent examples [38,40,50]. Microbial synthesis of nanoparticles is generally regarded as a response to toxic substances and typically follows a common pattern: metal ions are trapped within microbial cells or on the cell surface and subsequently reduced to form nanoparticles [3,51]. This reduction process, mediated by enzymes and capping agents, is classified into intracellular and extracellular synthesis, with comparative characteristics presented in Table 2.

Intracellular nanoparticle synthesis is more effective because of higher metal accumulation and greater nanoparticle production. However, extracellular synthesis is favored for its simpler purification process [50]. Some authors suggest that in algae, the extracellular pathway is more dominant than the intracellular one [46].

Four different techniques are used in the NP synthesis by photosynthetic microorganisms: the first uses living cells under their normal culture condition, the second method involves the use of the cyanobacteria and microalgae cells removed from the culture media and resuspended in the appropriate precursor solution, the third technique uses the cell-free culture media (supernatant) devoid of cells, and the fourth technique uses biomolecules extracted from cell biomass (Figure 7).

In vivo synthesis of nanoparticles utilizes living cyanobacterial or algal cultures in the log phase of growth, allowing limited interaction with the precursor solution. The internal biochemical reduction of metal ions to nanoparticles takes place in the cytoplasm, cell membrane, and thylakoid membrane [3]. Research reports suggest that bioreduction may be driven by enzymes like NADH-dependent nitrate reductase or by biomolecules such as polysaccharides, proteins, and pigments linked to metabolic processes like nitrogen fixation, photosynthesis, and cellular respiration [52]. Hydroxyl groups (–OH) of polysaccharides and pigments, as well as carboxyl groups (–COOH) from various amino acids, can bind metal ions and facilitate the formation of metallic nanoforms [53]. To illustrate microalgae-mediated biosynthesis of AuNPs in *Chromochloris zofingiensis*, Li X. and colleagues propose the following consecutive steps: 1—Au (III) is first trapped on the microalgal cell walls through electrostatic interactions with negatively charged cell wall components (such as –COOH, –OH, and –OSO_3_H groups from polysaccharides and proteins); 2—subsequently, Au(III) is likely bioreduced by –OH and –SH containing moieties present on the cell wall, as well as by reductase enzymes and electrons from the cytoplasm and thylakoids [54].

The in vitro green synthesis of nanoparticles depends on biomolecules that serve as reducing and capping agents [2]. The extracellular reduction of metal ions to their nanoform is possible because photosynthetic microorganisms secrete active reducing agents into the surrounding medium, such as polysaccharides, proteins, and pigments (including carotenoids, phycocyanin, and phycoerythrin), which influence nanoparticle nucleation and growth, leading to unique morphologies. The amino and thiol groups of proteins and sulfated polysaccharides also stabilize and coat metal nanoparticles in aqueous solutions [47,55].

Nanoparticle synthesis mediated by cyanobacterial and microalgal metabolites involves several distinct phases: an activation phase, during which metal ions are reduced, and nucleation occurs; a growth phase, characterized by the amalgamation of unit cells into crystallites; and a termination phase, where nanoparticles of various shapes and sizes become thermodynamically stable [56].

A study on metal NP synthesis using the cyanobacteria *Oscillatoria* sp. and *Spirulina platensis* examined three different methods: (1) cell-free supernatant (secondary metabolites), (2) cyanobacterial aqueous extract, and (3) whole-cell cultivation (in vivo). The characterization results confirmed the formation of silver oxide nanoparticles mediated by *Oscillatoria* sp. using cyanobacterial extract and whole-cell cultivation (intracellular synthesis), as well as the formation of gold nanoparticles mediated by *Spirulina* using cell-free supernatant and in vivo cultivation (extracellular synthesis) [57]. Mondal et al. reported that while microalgae commonly perform intracellular synthesis of iron nanoparticles via live-cell suspensions, macroalgae are usually used as cell-free extracts to promote extracellular nanoparticle formation [58].

The use of cyanobacterial and microalgal biomolecules as vectors for synthesizing metallic nanoparticles is supported by their effectiveness as capping and stabilizing agents, which inhibit nanoparticle overgrowth and prevent aggregation. These biomolecules can also modulate the shape, stability, and size of nanoparticles and play an essential role in reducing nanoparticle toxicity while enhancing their functional properties, biocompatibility, and bioavailability in living cells [46]. In their review of recent progress in algae-mediated silver nanoparticle synthesis, Choudhary S. and colleagues stated that the main advantage of the Ag NPs synthesized by algae is their wide variety in size and shape due to distinctive reducing/capping agents engaged in the synthesis process, notably functional groups in various biomolecules of proteins (–OH^−^), carbohydrates (–COO^−^, OH^−^, C–O–C), alkynes in lipids, secondary metabolites (alkaloids, flavonoids, polyphenols; –OH^−^, –C–O–C–, –C–O–, –C=C–, –C=O–), etc. [59]. FTIR analysis is commonly used to identify the functional groups involved in reducing metal salts to nanoparticles.

Therefore, scientists focus on approaches involving cyanobacteria and microalgae extracts, which contain cells’ total biochemical composition or single molecules, such as proteins, polysaccharides, lipids, or pigments. Numerous studies have reported the extracellular synthesis of spherical, octahedral, and cubic Au NPs using *Spirulina platensis*, highlighting the role of proteins and peptides as reducing agents [60,61,62]. Exopolysaccharides from *Chlorella* sp. show strong potential for green, single-step synthesis of polysaccharide-capped Au NPs, offering excellent stability across a broad pH range and high salinity conditions [63]. The synthesis and capping of silver nanoparticles were reported using a polysaccharide–protein matrix secreted by *Arthrospira platensis* in both normal and gamma-irradiated cultures [64]. In another study, carbon dots ranging from 1.2 nm to 11 nm (predominantly 2 nm to 6 nm) were successfully synthesized using carbohydrates and lipids extracted from the biomass of the microalgae *Acutodesmus obliquus*. Lipids are hypothesized to primarily function as capping agents [65]. Green microalga *Desmochloris edaphica*, extract, and cell-free medium were successfully used for obtaining Ag NPs of 77.9 and 62.7 nm with a surface charge of −24.4 and −25.8 mV, respectively. According to the authors, the negative surface charge of the Ag NPs likely results from the algal corona, which coats the particles with functional groups such as O–H, C–H, C=C/N–H, C–O, C=O, and C–Br. This suggests that algal polysaccharides and proteins serve as reducing agents, while polysaccharides and fatty acids function as stabilizers during nanoparticle formation. The most abundant biomolecules detected by gas chromatography–mass spectrometry in the algal extract were elaidic acid (18.36%) and monoolein (17.37%), suggesting that these fatty acids may contribute to the stabilization of Ag NPs [66]. Fe_3_O_4_ nanoparticles were phyco-synthesized using a 0.1 M ferric/ferrous chloride solution (2:1 ratio) at 65 °C with an aqueous extract of the green microalga *Chlorella* K01. FTIR analysis revealed that the combined action of microalgal proteins, carbohydrates, and polyphenols facilitated the biosynthesis of spheroid Fe_3_O_4_ NPs with an average diameter of 76.5 nm [67].

Pigments are among the most promising microalgal and cyanobacterial biomolecules for metal nanoparticle synthesis. Phycobiliproteins (phycocyanin, allophycocyanin, and phycoerythrin), which are unique to cyanobacteria, have demonstrated efficacy in multiple studies, as highlighted in a recent review by Koli et al. focused on pigment-mediated nanoparticle synthesis [68]. For example, phycobiliprotein extract in a ratio 9:1 *v*/*v* to AgNO_3_, at room temperature and pH 7, facilitated the synthesis of spherical Ag NPs by *Spirulina platensis* and *Nostoc linckia*, with average particle sizes of 21.05 nm and 20.48 nm, respectively. Spirulina phycobiliproteins completed nanosynthesis in 24 h, while Nostoc phycobiliproteins completed it in 96 h [69]. The mechanism of phycobiliprotein involvement in nanosynthesis is not entirely understood. Still, it is known that their negatively charged functional groups (e.g., hydroxyl, carboxyl) can facilitate the binding of cationic ions.

The rate of nanoparticle production, as well as their quantity, shape, quality, and stability, depends on various parameters such as the species of cyanobacteria or microalgae, biomass or extract concentration, metal precursor concentration, pH, reaction time, temperature, illumination, and culture conditions [39,70,71,72]. A single factor does not govern nanosynthesis; multiple parameters collectively influence the resulting nanoparticles’ synthesis rate, properties, and functionality [73]. Controlled morphology and stabilization of biosynthesized nanoparticles can be achieved by optimizing reaction parameters and carefully selecting the appropriate microbial strain [74].

The database analysis performed by Lamila-Tamayo et al. presents relevant cyanobacteria and algae species involved in nanosynthesis, the effects of various factors influencing this process, the predominant types of synthesized NPs, and their possible applications. According to this database analysis, eukaryotic microalgae were the most frequently studied for nanoparticle synthesis, with dried algal powder and whole living cultures being the most commonly used approaches. The most used species in nanosynthesis were the microalgae *Chlorella vulgaris*, *Chlamydomonas reinhardtii*, *Euglena gracilis*, *Dunaliella salina*, and the cyanobacteria *Spirulina platensis*. The authors noted a consistent trend across studies, regardless of algal type: most focused on synthesizing silver nanoparticles (about 50%), followed by gold nanoparticles at a significantly lower rate (19.17%) [75].

Many research reports have demonstrated the possibility of directing bio-nanosynthesis for specific applications by controlling cultivation and reaction parameters. For instance, nanosynthesis is directly influenced by biomass/extract concentration and precursor concentration. The yield of nanoparticles increases proportionally with biomass/extract concentration until reaching a threshold, beyond which nanoparticle agglomeration occurs. It is well established that precursor concentration affects yield and plays a key role in shaping the morphology of synthesized nanoparticles [46].

The incubation period is an important factor in bio-nanosynthesis and should be optimized based on the biological system and desired nanoparticle characteristics. The studies in the present review report a wide range of interaction times between reducing agents and metal ions (with extremes of 5 min and 1 month). Generally, most studies confirm the optimal incubation of 24–48 h. Still, the optimal time varies depending on the biological source, synthesis techniques, metal precursor, reaction conditions, and the purpose of synthesis. Higher temperatures accelerate the NP synthesis and reduce the required incubation time. In vivo synthesis requires a more extended incubation period (24–72 h), while extracts complete the nanosynthesis faster (30 min–24 h) [57,59,76]. In determining the optimal incubation time, researchers consider the final application of the resulting nanoparticles. For instance, a study on the biochemical changes in the microalga *Porphyridium cruentum* linked to silver nanoparticle biosynthesis suggests that although optimal nanosynthesis was achieved after 24 h of incubation, the incubation time should be reduced to prevent biomass destruction when aiming to obtain biomass enriched with silver nanoparticles [77].

The pH level significantly affects the size, shape, texture, stability, and localization (extracellular or intracellular) of the produced nanoparticles. As mentioned in scientific papers, at neutral pH, Au NP biosynthesis primarily occurs in the periplasmic region of cyanobacteria. It exhibits greater variation in size and morphology at the more acidic pH. At pH 2.0, small Au NPs are deposited on bacterial cell walls, whereas larger particles are found in the extracellular matrix [78]. In fact, at acidic pH, a high concentration of H^+^ in the solution decreases the reduction potential of extracellular functional groups, leading to particle agglomeration and the formation of larger nanoparticles. In contrast, alkaline pH enhances the reduction power of surface functional groups, preventing particle agglomeration and producing smaller nanoparticles. The role of intramolecular hydrophobic and hydrophilic interactions is also taken into account [79]. The pH of the reaction mixture can affect the reducing activity of biomolecules involved in NP biosynthesis. In acidic environments, these biomolecules may undergo protonation, altering their ability to interact with metal ions, which can reduce synthesis efficiency. In alkaline conditions, deprotonation enhances the binding of metal ions to biomolecules, improving NP stabilization and preventing aggregation [2]. Generally, a slightly alkaline pH (7.5–9.0) is favorable for the synthesis of well-dispersed, stable nanoparticles, but specific pH conditions vary depending on the type of metal and microalgal species used. In a study on the use of the microalga *Asterarcys* sp. for the biological reduction of AgNO_3_ to synthesize silver nanoparticles, the optimal conditions were identified as an AgNO_3_ concentration of 3 mM, an incubation duration of 24 h, and a pH of 9 [80]. To obtain titanium NPs using the supernatant of *Phaeodactylum tricornutum*, the optimal conditions were applied: incubation for 1 h and pH 7.5 [81].

Temperature is another important thermodynamic variable in the synthesis of nanomaterials. As temperature rises, particles gain kinetic energy, move faster, and accelerate the reaction rate. Physical and chemical nanosynthesis typically involves high temperatures (>350 °C). However, cyanobacteria- and microalgae-mediated nanosynthesis is performed at temperatures ranging from 20 °C to 40 °C in the case of in vivo synthesis and from 50 °C to 100 °C, with extremes at room temperature and 200 °C in the case of in vitro nanosynthesis, as reported in the studies analyzed in this section of our review. For example, in vivo synthesis of Ag_2_O/AgO NPs by *Oscillatoria* sp. and AuNPs by *Spirulina platensis* was reported during their cultivation at room temperature, pH 9, and under illumination, using AgNO_3_ and HAuCl_4_ as precursors, respectively. However, when using cell-free media and aqueous extracts of the same cyanobacterial strains, different optimal conditions were required to synthesize the same NPs: a temperature of 50 °C and an incubation period of 30 min [57]. Algal extract of *Coelastrella terrestris* was successfully used for the phyco-fabrication of silver nanoparticles under the following optimum synthesis conditions: pH 10, room temperature, and 24 h incubation time [82]. To synthesize Au NPs, the algal extract of *Phormidesmis communis* was mixed with 0.5 mM of precursor HAuCl_4_ (ratio 2:1 *v*/*v*) for 30 min at 100 °C and pH 5.6 [76]. Using microalgae as the sole precursor and adjusting only temperature and pH, Bi L. and colleagues [83] produced rattle-type microspheres with controllable mesoporous outer shells (19.4–46.3 nm) and multicomponent nanocores (Ca_5_(PO_4_)_3_OH and Fe_3_O_4_/MgFe_2_O_4_). For the fabrication of microspheres, *Chlorella pyrenoidosa* powder was dispersed in 40 mL NaOH solution (pH 14) and stirred intensely at 100 °C for 2 h [83]. The broad temperature range used in cyanobacteria- and microalgae-mediated nanosynthesis likely reflects variations in biochemical composition and heat sensitivity among different organisms. Recent studies have investigated how light exposure affects microalgae-mediated nanoparticle synthesis, revealing its significant role in influencing both the production rate and characteristics of the synthesized nanoparticles. Regarding the photosynthetic activity of cyanobacteria and microalgae, light conditions are particularly important during in vivo NP synthesis and play a crucial role in all techniques, depending on the type of nanoparticles synthesized, the synthesis method, and the microorganism used. For example, in the extracellular synthesis of AgNPs using *Planophyla laetevirens* biomass extract and cell-free medium, the optimal condition for nanoparticle formation in the cell-free medium was 1 h in the dark at 80 °C—since light exposure increased particle size and agglomeration. However, with the algal biomass extract, silver nitrate was not reduced to AgNPs in the dark, making light incubation an optimal condition for synthesis. The authors suggest that the bioreduction of silver nitrate by algal extracts is a photo-reduction process where light plays a key role [84]. The extract of *Nostoc muscorum* biomass required the following optimal conditions to convert silver nitrate to Ag NPs: 25 °C, under light illumination, for 24 h, at pH 7.4 [85]. *Desmochloris edaphica* extract and cell-free medium were used for the biosynthesis of AgNPs under established optimal conditions. For the microalgal supernatant, 1 mL of algal extract was mixed with 9 mL of 2 mM AgNO_3_ and incubated at 25 °C under illumination at pH 7 for 24 h. For the cell-free supernatant, 1 mL was mixed with 2 mL of 2 mM AgNO_3_ and incubated at 60 °C for 1 h, followed by incubation under illumination at pH 9.5 for 24 h [66]. Spirulina biomass was exposed to a 0.05 M FeCl_3_ solution at pH 5.5 in the dark for 48 h, resulting in the biosynthesis of spindle-shaped nano-iron particles measuring 24.77 ± 5.26 nm [86]. Silver nanoparticles were synthesized using the cell-free supernatant of a *Haematococcus pluvialis* culture under illumination at 45 °C and pH 11.0. AgNO_3_ concentrations from 1 to 5 mM served as the limiting reactant. Synthesis occurred under both static and stirred conditions; continuous stirring sped up the reaction but caused aggregation during extended incubation [87].

An original approach to the de novo manipulation of nanobiosynthesis was adopted by Mohamed and colleagues by studying the effect of five phytohormones—indole acetic acid, kinetin, gibberellic acid, abscisic acid, and methyl jasmonate—on Ag NP production by the cyanobacterium *Cyanothece* sp. After 60 days of cultivation, harvested biomass was incubated with 0.1 mM AgNO_3_ and varying concentrations of these phytohormones. The cyanobacterium produced spherical Ag NPs sized 70 to 140 nm. Indole acetic acid and kinetin significantly boosted Ag^+^ to Ag^0^ conversion rates (87.29% and 55.16%), while gibberellic acid and abscisic acid had little effect (45.23% and 47.95%) compared to the control. Methyl jasmonate increased conversion to 90.29%. The authors concluded that phytohormones can modulate green Ag NP synthesis in *Cyanothece* sp., offering potential applications in agriculture and biomedicine [88].

As demonstrated by the examples above, researchers optimize the nanosynthesis process by refining key parameters while exploring new strains with potential applications in nanotechnology. Special attention has recently been focused on the cyanobacterium *Desertifilum*, with new strains being proposed for the nanosynthesis of different NPs. A novel cyanobacterial strain, *Desertifilum* IPPAS B-1220, was isolated, purified, cultured, genetically identified, and studied for AgNP synthesis. Its cell filtrate enabled the green synthesis of spherical AgNPs measuring 4.5–26 nm in diameter at room temperature under direct illumination (2000 ± 200 lux) for 24 h [89]. Another newly characterized strain, *Desertifilum* sp. EAZ03 has been proposed for the biosynthesis of zinc oxide nanoparticles. Its cell extract effectively mediated the formation of rod-shaped ZnO NPs with an average size of 88 nm [90]. In another study, cell-free supernatants and biomass extracts from *Phormidium ambiguum* and the novel *Desertifilum tharense* efficiently produced silver nanoparticles under both light and dark conditions. The nanoparticles ranged in size from 6.24 to 11.4 nm and 6.46 to 12.2 nm, respectively, with the best results obtained using culture supernatants, especially under light [91].

The list of new strains applied for bio-nanosynthesis can be expanded to include the cyanobacterial strain *Desmonostoc alborizicum*, used for the first time in the synthesis of Se nanoparticles [92]; the cyanobacterium *Nodosilinea nodulosa* for the synthesis of Co_3_O_4_ nanoparticles [93]; and the green alga *Botryococcus braunii* as a reducing, stabilizing, and capping agent to synthesize stabilized nanoscale palladium and platinum nanoparticles [94].

### 3.3. Cyanobacteria- and Microalgae-Based Nanosynthesis for Agriculture

Nanotechnology has revolutionized agriculture and related fields. Agro-nanotechnology is recognized as an innovative strategy for sustainable agriculture, contributing to soil quality improvement, smart fertilization, precision farming, crop production under stress, remediation of contaminated soil and water, agro-waste management, and sustainable energy generation [95,96,97].

Recent publications highlight the potential benefits of nanomaterials—particularly nanoparticles—in a wide range of agricultural applications. These include, but are not limited to, nano-fertilizers and nano-biostimulants, nano-herbicides, nano-fungicides, nano-pesticides, nano-enzymes, nano-biosensors, nanoscale genetic carriers, nano-bioremediation agents, and nanocomposites for packaging. The agricultural applications of nanomaterials identified in the analyzed publications are summarized in Table 3.

Given the recent trends in nanotechnology applications, many researchers have focused on the biological synthesis of nanomaterials for sustainable agricultural practices [115,116,117,118,119,120].

Using plant, animal, and fisheries wastes to synthesize metallic and non-metallic nanoparticles is proposed as an innovative method for circular bioresource utilization in climate-smart and stress-resilient agriculture [121,122].

As noted by Xu L. and colleagues, bioinspired synthesis offers several advantages over chemical synthesis: (a) it is a facile, one-pot process, as bioactive substances act as both reducing and capping agents; (b) it is cost-effective and scalable, relying on inexpensive raw materials and simple procedures; (c) it enables nanomaterial functionalization, improving stability and effectiveness for various applications; and (d) it eliminates the need for toxic or hazardous chemicals, enhancing the biocompatibility of the final product [8].

According to numerous authors, the biosynthesis of nanomaterials using bacteria, algae, yeast, fungi, actinomycetes, and plants has opened new avenues for producing inorganic nanoparticles as environmentally friendly fertilizers. The term “nanobiofertilizer” refers to the intentional integration of biocompatible nanomaterials with biologically derived fertilizers, demonstrating strong efficacy. The application of nanobiofertilizers preserves soil nutrient content and enhances crop growth and productivity by activating various physiological and biochemical processes [123,124,125].

“Nano biostimulants” are an emerging technology that promotes plant growth and development by stimulating biological processes rather than directly supplying nutrients [126]. For instance, foliar spraying of green-synthesized silver nanoparticles on NaCl-stressed pearl millet seedlings enhanced osmotic regulation and O_2_–H_2_O_2_ scavenging mechanisms, helping to mitigate oxidative stress damage and improve growth. According to the study’s authors, the application of Ag NPs positively influenced growth metrics, chlorophyll content, osmolytes, and antioxidant mechanisms in salt-stressed seedlings, with effects that showed a dose-related increase [127]. A study investigating the morphophysiological responses of barley (*Hordeum vulgare* L.) to biologically synthesized silver nanoparticles—produced using an extract from the medicinal plant *Ochradenus arabicus*—demonstrated significant enhancements in growth performance, chlorophyll content, and enzymatic activity following AgNP treatments [128]. In another study by the same authors, copper nanoparticles biosynthesized using *Solenostemma argel* were applied to barley plants under salinity stress. The treatment with biologically synthesized CuNPs alleviated the adverse effects of NaCl by promoting plant growth, enhancing gas exchange parameters, increasing photosynthetic pigment content, and improving osmoregulation. It also reduced the accumulation of MDA and H_2_O_2_. Additionally, protection against NaCl-induced stress was achieved by lowering total phenol and flavonoid levels and increasing the activity of antioxidant enzymes [129].

In addition to the use of plant extracts, which are remarkably involved in NP synthesis, scientists have started paying attention to microbial nanosynthesis. Various microbial resources—including microalgae, fungi, actinomycetes, bacteria, viruses, and secondary microbial metabolites—are employed in nanoparticle synthesis. The resulting nanoparticles exhibit unique biocompatibility, broad applicability, cost-effective production methods, and environmental sustainability [96,110,130,131,132,133].

The analysis of scientific publications selected for our review highlights the tremendous potential of cyanobacteria and microalgae in enhancing agricultural production while preserving the environment. As discussed above, the ability of photosynthetic microorganisms to produce nanoparticles with controlled size, shape, and distribution enhances the biocompatibility of NMs, which is essential for their specific applications.

Various innovative strategies are used to implement agricultural applications of bionanomaterials for plant growth, protection, and stress adaptation (Figure 8). The following sections explore key bionanotechnological aspects of cyanobacteria and microalgae applications in agriculture, which are systematically presented in Table 4.

#### 3.3.1. Application in the Agri-Food Sector of Nanoparticles Synthesized by Photosynthetic Microorganisms

Green synthesis of metal nanoparticles using cyanobacteria and microalgae holds significant promise for enhancing global food production and safety, particularly in developing nano-pesticides, nano-biosensors, and nano-fertilizers [8,86,105,145,146]. Concerning the utilization of cyanobacteria- and algae-based NPs in agriculture, they can be applied as nanoparticles alone or as nanocomposites.

Many studies on using NPs produced by cyanobacteria and microalgae in agriculture focus on their effectiveness as pest management agents [145,147,148]. As mentioned by El-Refaey and Salem, metallic and metal oxide nanoparticles (copper, magnesium oxide, gold, selenium, iron oxide, palladium, zinc oxide, titanium dioxide, and silver) in their biosynthesized form, besides their well-known role in protecting plants from bacterial, fungal, and pest infections, can also act as plant growth promoters, enhancing crop yield [148]. A range of studies has demonstrated this dual effect of biogenic NPs, as illustrated in the following examples.

Iron oxide nanoparticles (Fe_3_O_4_ NPs), synthesized and stabilized using an aqueous extract of the green alga *Chlorella* K01, exhibited a significant stimulatory effect on the germination and vigor index of rice, maize, mustard, green gram, and watermelon, while also demonstrating antifungal activity against *Fusarium oxysporum*, *Fusarium tricinctum*, *Fusarium moniliforme*, *Rhizoctonia solani*, and *Pythium* sp. [67].

Silver nanoparticles have been evaluated as efficient, eco-friendly alternatives to bactericides, fungicides, and nematicides in both in vitro and greenhouse studies [9]. These studies demonstrate that AgNPs can disrupt membrane integrity, induce oxidative stress in phytopathogens, and modulate protein expression and metabolic profiles in host plants.

The biogenic synthesis of Ag NPs using a cell-free extract of *Leptolyngbya* sp. WUC 59, a cyanobacterium isolated from polluted water, is proposed as a clean and sustainable tool for controlling bacterial diseases in agricultural crops and promoting seed germination and early growth of wheat (*Triticum aestivum* L.) [149].

A further study discusses the production of Ag NPs using soluble polysaccharides extracted from *Chlorella vulgaris* and their biostimulatory activity. The biosynthesized NPs and their antimicrobial properties exhibited a stimulatory effect on plant growth when used for seed priming in *Triticum vulgare* and *Phaseolus vulgaris*. This effect was expressed through increased root length, leaf area, and shoot length and enhanced protein, carbohydrate, and photosynthetic pigment content [134].

Antifungal screening of SeNPs biosynthesized by microcystin-producing *Desmonostoc alborizicum* revealed strong antifungal properties, with *Alternaria alternata* showing the highest sensitivity (9.66 ± 0.51 µg/mL) [92].

An extract of the cyanobacterium *Nodosilinea nodulosa* served in the synthesis of cobalt oxide nanoparticles (Co_3_O_4_ NPs) with a size of 41 nm. Their antifungal activity was evaluated against *Aspergillus flavus* and *Fusarium oxysporum*, with the latter exhibiting the highest vulnerability. However, the obtained Co_3_O_4_ NPs were inefficient against the insects *Tribolium castaneum*, *Sitophilus oryzae*, and *Rhyzopertha dominica* [93].

The organic biomass extract of *Nostoc* sp. SI-SN offers a safe, sustainable, and green method for producing multifunctional ZnO nanoparticles for biomedical uses and as nano-fertilizers in agriculture. Supporting this, ZnO NPs at 10 and 15 µg/mL significantly improved seed germination, seedling growth, and chlorophyll content in corn and wheat, respectively [135].

According to Abideen and colleagues, algal biomass-derived products, including NPs, have shown the potential to mitigate abiotic stresses like drought, salinity, and flooding and to aid phytoremediation of toxic elements in agricultural soils [150].

In their review, Sowmiya et al. present cyanobacteria and microalgae as key components in nanobiofertilizer production and valuable tools for enhancing plant growth and nutrient uptake (e.g., *Anabaena cylindrica* enhances nodulation, plant growth, and yield in common beans; *Chlorella* sp. accelerates seed germination, root and leaf development, and photosynthesis in maize), increasing stress tolerance (*Scytonema hofmanni* and *Dunaliella salina* exhibit high salinity tolerance), and improving soil quality (*Oscillatoria* sp., *Microcoleus vaginatus*, and *Nostoc commune* secrete extracellular polymeric compounds that stabilize soil, prevent erosion, and support biological crust formation) [151].

Agricultural nanotechnology leverages the synergy of nanoparticles and the biomass of cyanobacteria and microalgae to enhance efficiency and sustainability.

#### 3.3.2. Nanoparticle-Enriched Extracts of Cyanobacteria and Microalgae in Plant Biotechnology

One of the innovative and sustainable agricultural strategies with great potential to replace traditional agricultural practices explores the synergies between nanomaterials and photosynthetic microorganisms. The synergistic effect of co-delivery systems based on nanoparticles can be successfully utilized in pest control and in the development of new biofertilizers. Integrating biofertilizers with nanoparticles to enhance plant growth is defined as “nano biofertilisers”. Cyanobacteria, microalgae, and Azotobacter populations are the best candidates for biostimulants and plant biofertilizers [152]. The example of nitrogen-fixing cyanobacteria is particularly significant. Utilizing the symbiotic cyanobacteria-plant interaction and metallic NPs as carriers of essential nutrients can enhance nutrient transfer and bioavailability and modulate plant stress response and disease resistance [153].

A detached leaf assay was used to evaluate the biocontrol efficacy of silver nanoparticles (AgNPs), alone and combined with *Calothrix elenkinii*, against *Alternaria alternata* infection in tomato. Treated leaves showed higher chlorophyll levels and reduced endoglucanase activity compared to infected controls. Microscopic analysis revealed that AgNPs augmented with cyanobacteria effectively inhibited pathogen growth, with the combined treatment demonstrating greater stability over time. This approach provided valuable insights into the interactions between cyanobacteria, nanoparticles, and pathogen-infected leaves, highlighting the potential of AgNPs combined with *Calothrix elenkinii* as a scalable biocontrol strategy [136].

The synergistic biocontrol effects of silver nanoparticles augmented with *Calothrix elenkinii* against *Alternaria alternata* were confirmed through foliar application on infected tomato plants. Disease severity was reduced by 47–58%, while leaf chlorophyll, carotenoid content, and polyphenol oxidase activity increased significantly by 44–45%, 40–46%, and 23–33%, respectively. Ergosterol content decreased by 63–79%. Enhanced antioxidant enzyme expression, improved plant growth, and reduced fungal load demonstrate the strong biocontrol potential of *Calothrix elenkinii*-augmented AgNPs. This synergistic approach offers a promising strategy to manage *A. alternata* infection in tomato under various agroclimatic conditions [137].

French basil was foliar-treated with copper nanoparticles (CuNPs) and *Spirulina* sp. extract synergistically. Plants treated with 500 mg/L CuNPs and 1.5 g/L *Spirulina* extract showed the best growth, oil production, and highest chlorophyll and carotenoid levels. While CuNPs alone significantly boosted antioxidant activity, their combination with *Spirulina* extract reduced antioxidant levels, suggesting a protective effect from the cyanobacterial stimulant [138].

Seed priming in triticale using extracts from *Arthrospira platensis* cultivated with copper nanoparticles and copper oxide nanoparticles resulted in significant increases in chlorophyll, carotenoid, and phenolic content and enhanced antioxidant activity in the leaves. Treatment with nanoparticle-enriched extracts also reduced malondialdehyde (MDA) levels, indicating a protective effect against oxidative stress [139].

El-Semary and colleagues investigated the mitigation of salinity stress on crops using microbial inoculants combined with nanomaterials and methyl salicylate. Seeds exposed to varying salinity levels were treated with microbial formulations (*Cyanothece* sp. and *Enterobacter cloacae*), alone or combined, with or without methyl salicylate. Nanomaterials—graphene, graphene oxide, and carbon nanotubes—were also applied with biofertilizers at the highest salinity. Treatments were delivered via soil and capsules. Salinity stress, especially at −5 MPa, severely inhibited growth, but microbial treatments, particularly combined with methyl salicylate, significantly mitigated these effects. The inclusion of nanomaterials with biofertilizers further reduced the inhibitory impact of salinity. This approach suggests that both encapsulated and in-soil synergistic biofertilizers and nanomaterials can effectively mitigate crop salinity stress [140].

A recent study focused on the effect of the microalgal extract of *Scenedesmus obliquus* and its biosynthesized zinc oxide nanoparticles as a foliar application on tomato growth and yield during the late summer season. The microalga *Scenedesmus obliquus* was used for the green synthesis of zinc nanoparticles, and it was applied along with microalgal extract as a foliar fertilizer to boost tomato growth and yield. Treatment with algal extract plus 250 ppm bio-NPs led to a 37% increase in total fruit yield and a 43.1% rise in marketable yield. Soil quality also improved, with total microbial activity up 369.4% and dehydrogenase activity up 298.8%. Zinc accumulation increased by 74.8%, 182.7%, and 104.2% in roots, leaves, and fruit, respectively, compared to controls. The authors suggest that phyco-synthesized ZnO NPs stimulate meristematic cell activity by activating key biochemical pathways, enhancing tomato biomass accumulation [141].

Another current study aimed to propose a nano-iron phycofertilizer that improves crop quality in an eco-friendly way. Spindle-shaped nano-iron particles of 24.77 ± 5.264 nm were obtained using *Spirulina* sp. biomass. Field trials compared different fertilizers, including phyco-synthesized nano-iron (INP), dried *Spirulina* sp. biomass (SB), and nano-iron-loaded *Spirulina* sp. biomass (NPF), against conventional NPK fertilizer. NPF showed the best results, increasing rice shoot length by 37.6%, crop productivity by 47%, and grain weight by 15% compared to NPK. Additionally, NPF-treated rice grains had 44% and 28% higher iron content in unpolished and polished forms, respectively. A cost-effectiveness analysis highlighted NPF as a superior, eco-friendly alternative to conventional fertilizers [86].

The effects of silver nanoparticles (AgNPs) and *Chlorella* sp. MF1 suspension on the vegetative growth of *Eruca sativa* were evaluated through seed soaking and foliar spraying at varying concentrations. Results indicate that AgNPs and *Chlorella* sp. MF1 act as effective biostimulants, enhancing the growth of *Eruca sativa* [142].

To improve tomato plant performance, a novel bio-stimulant was developed using chitosan nanoparticles and microalgae-based protein hydrolysate from *Arthrospira platensis*. Two lecithin-/chitosan-based nanoparticle formulations (NP) were created to deliver the protein hydrolysate (PH) containing 68.9 mg/mL of free amino acids. The study tested the effects of these formulations on tomato plant growth by applying weekly sprays of PH, its nanoparticle derivatives, and a control. Results showed that both NP and PH treatments significantly stimulated plant growth, up to a 49.5% increase in plant height during the vegetative phase. During the fruiting phase, NP treatments (0.3 and 0.8 mg/mL) optimized the production of chlorophyll, carotenoids, flavonoids, epigallocatechin, and lycopene in fruits. These findings suggest that the developed nanoparticle bio-stimulant effectively enhances growth, nitrogen fixation, and antioxidant metabolite production in tomatoes, with potential applications for sustainable agriculture [143]. One emerging research approach for inhibiting ecosystem desertification is the combination of mixed cyanobacteria with nanocomposites to generate biocrusts. By combining the advantages of cyanobacteria as a low-cost source of water, nitrogen, and phosphorus with the metal–organic framework of carboxymethyl cellulose, newly developed network-structured nanocomposites, with their high specific surface area and diverse surface functionalities, demonstrate excellent water and nutrient retention along with strong biosafety [154].

#### 3.3.3. Cyanobacteria- and Microalgae-Based Nanosensors

Photosynthetic microorganisms, due to their quick responsiveness, short regeneration time, and ease of maintenance, are known to be bioreceptors in developing biosensors—devices that translate biological responses into electrical signals for monitoring environmental quality. Using whole microalgal cells, enzymes, thylakoid membranes, and biomass in developing biosensors for the optical detection of pesticides and herbicides presents a cost-effective, environmentally friendly, and feasible strategy with environmental benefits [155,156,157].

Therefore, researchers consider these organisms the perfect platform for developing a new generation of biosensors, namely nano-biosensors. Nanomaterials such as carbon nanotubes, graphene, and nanospheres have proven effective in developing biosensors with enhanced sensitivity and performance. For instance, cellulose/inorganic composite rattle-type microspheres with tunable mesoporous shells and multicomponent nano-cores have been developed by adjusting only pH and temperature using *Chlorella pyrenoidosa* biomass as a single-source precursor. Their enhanced sensitivity stems from tailored pore size, void space, and natural carboxyl groups, which facilitate enzyme loading and bioconjugation. The magnetite and hydroxyapatite cores enable magnetic separation and signal amplification. Thus, the authors propose a new eco-conscious strategy for elaborating advanced sensors for pollutant detection [83].

To improve sensitivity and performance, recent research focuses on immobilization techniques, such as adsorption, entrapment, encapsulation and covalent bonding, using nanomaterials [155]. Conversely, the microalgal biosensors proved an adequate analytical tool for assessing sustainable nano-herbicides used in smart agriculture. For instance, a novel, eco-friendly biosensor was designed by Antonacci and colleagues to detect biodegradable nanoparticle-encapsulated herbicides and promote sustainable agriculture. The biosensor uses whole cells of the *Chlamydomonas reinhardtii* UV180 mutant immobilized on carbonized lignin electrodes and integrated into a photo-electrochemical transducer for detecting nano-formulated atrazine, encapsulated into zein- and chitosan-doped poly-ε-caprolactone nanoparticles (atrazine–zein and atrazine-PCL-Ch). The biosensor demonstrated a linear response for atrazine detection in the 0.1–5 μM range, with detection limits of 0.9 nM and 1.1 nM for atrazine–zein and atrazine-PCL-Ch nanoparticles, respectively. It showed no interference from common pollutants and no matrix effect in wastewater samples, with excellent recovery values. The biosensor exhibited a stable performance for up to 10 h [144]. Precision farming utilizes nano-sensors as a crucial tool to predict the occurrence of soil diseases and monitor bacterial activity in the environment. This ensures the efficient use of natural agricultural resources such as water, herbicides, and nutrients [158]. However, further research and development are required to effectively apply nano-biosensors in agriculture.

#### 3.3.4. Other Nanotechnological Applications of Cyanobacteria and Microalgae Impacting the Agri-Food Sector

Cyanobacteria’s and microalgae’s roles in nanotechnology extend beyond the biosynthesis of metal nanoparticles. They also serve as valuable tools for developing advanced nanotechnological systems, such as scaffolds and nanocoatings, designed to carry and protect bioactive compounds for various biotechnological applications. Recent scientific publications discuss strategies for nanoencapsulation of cyanobacterial and microalgal proteins, bioactive peptides, pigments, and polyunsaturated fatty acid for developing scaffolds using cyanobacteria and microalgae biomass and biocomponents, as well as the incorporation of microalgal compounds into nanospheres, nanocapsules, and nanoemulsions for smart food packaging [159,160,161]. The use of genetically engineered strains in nanotechnology is analyzed as another emerging trend in future applications of photosynthetic microorganisms [162].

An original study proposes the intracellular biosynthesis of gold nanoparticles as a tool for monitoring microalgal biomass by surface-enhanced Raman spectroscopy (SERS), a powerful technique for analyzing biomass composition but typically requires pre-synthesized Raman substrates. This study introduces an intracellular biosynthesis method for Au NPs as SERS substrates via biomineralization of HAuCl_4_ in *Chromochloris zofingiensis*. Gold nanospheres (20 nm) and hyperbranched nanostructures (120 nm) formed inside the cells without external reducing or capping agents. The gold hyperbranched nanostructures enhanced the Raman signal 9.7 times, particularly for carotenoids, while maintaining biomass yield. This method was also applied to other microalgae species, demonstrating its potential as a rapid and effective strategy for real-time monitoring of microalgal biomass composition [154].

Researchers are also focusing on the role of nanomaterials in microbial metabolism and their interaction with photosynthetic microorganisms. Many studies evaluate the effect of engineered nanomaterials on the growth rate, biochemical changes, ultra-structural changes, and nanoparticle toxicity mechanisms in cyanobacteria and microalgae. The relevance of these studies is determined by the necessity to assess the impact of nanomaterials on living organisms, a major global concern.

Cherchi and colleagues assessed the impact of nano-titanium dioxide (nTiO_2_) on nitrogen metabolism in the nitrogen-fixing cyanobacterium *Anabaena* PCC 7120 by analyzing transcriptional changes in biomarker genes related to nitrogen regulation, fixation, assimilation, and storage. Their findings suggest that nTiO_2_-induced environmental stress leads cyanobacteria to change their intracellular carbon–nitrogen balance, potentially affecting broader ecological trophic interactions [163]. According to Kumar et al., the main causes of nanoparticle toxicity to cyanobacteria include shading on the cell surface that limits light, generation of reactive oxygen species activating antioxidant defenses, pigment degradation, and membrane damage [164].

Comparing the toxicity of tungsten and vanadium oxide nanoparticles on *Spirulina platensis*, scientists found that vanadium oxide nanoparticles (VNPs) were absorbed more and proved toxic to the cyanobacteria. In contrast, tungsten nanoparticles (WNPs) at all tested concentrations (0.001, 0.017, and 0.05 g/L) enhanced protein and phycocyanin content in *Spirulina* [165]. In another study, silver nanoparticles (1–50 mg/L exhibited a dose-dependent toxic effect in *Nannochloropsis oculata*, manifested by significant growth inhibition, decreased chlorophyll content, increased enzymatic and non-enzymatic (carotenoid) antioxidant activities, and elevated LDH activity. The results showed that AgNPs can increase the fatty acid content in *N. oculata*, which can be regarded as a defense mechanism against NP toxicity [166].

The increased lipid synthesis and altered fatty acid composition in *Chlorella* sp. UJ-3 induced by oxidative stress caused by exposure to iron nanoparticles (NPs) was explored by Wang and colleagues. Exposure to low concentrations of Fe_3_O_4_ nanoparticles (20 mg/L) significantly increased algal biomass, while the highest total lipid content occurred at a higher concentration (100 mg/L). The proposed strategy involves initially treating microalgae with low Fe_3_O_4_ NP levels, followed by a higher dose after 12 days. This approach boosted biomass by 50% and total lipid production by 108.7% compared to untreated controls [167].

Yuan et al. review the use of nanotechnology to enhance microalgae production of high-value metabolites. Nanomaterials are proposed as novel and supporting solutions for the existing production of metabolites, such as lipids, proteins, pigments, and exopolysaccharides. Additionally, nanotechnology treatment can induce structural and functional modifications in microalgal metabolites, particularly polysaccharides. The primary mechanisms by which nanomaterials boost algal growth and metabolite synthesis include enhancing photosynthesis and controlling reactive oxygen species levels [168].

Recent studies have revealed beneficial microbe–nanoparticle interactions and the utilization of naturally occurring nanoparticles by microbes as functional “nano tools” assisting in electron transfer, chemotaxis, and substrate storage. In several species of cyanobacteria, naturally formed nanoscale carbonate inclusions (50–870 nm) have been discovered, concentrated at the cell pole, septum, or throughout the cytoplasm. The possible roles of these inclusions in cellular processes could include involvement in the Ca^2+^ regulation system during cell division, regulation of cell density, or pH control [169]. In addition, it has been found that nanoparticles have applications in gene expression regulation in cyanobacteria and microalgae, as well as in enhancing carbon dioxide fixation and enabling bioprocess production in photosynthetic microorganisms [170]. This makes nanotechnology an outstanding tool for improving, optimizing, and exploiting valuable compounds from cyanobacteria and microalgae.

## 4. Conclusions, Future Directions, and Final Remarks

Nanotechnology and biotechnology—two dynamic fields of science and technology—join their efforts to become key components in sustainable agricultural development. With mounting pressures from population growth, climate change, and environmental degradation, there is a growing need for innovative solutions that enhance productivity while preserving ecological balance. In this context, nanotechnology has attracted considerable attention for its potential to reduce the negative impacts of excessive pesticide and fertilizer use, improve crop yields, and mitigate environmental pollution. Nanomaterials, through their capacity for controlled release and enhanced nutrient delivery, offer a more efficient and targeted approach to agricultural management. To understand the behavior of nanoparticles in plants and promote their targeted use in agriculture, recent research has focused on nanobionics—an emerging field at the intersection of nanotechnology and biology—that integrates nanostructured materials with biological systems [171,172,173].

A particularly promising development is the biosynthesis of nanoparticles using living organisms, which represents a novel and environmentally friendly frontier in biotechnology [174,175]. While plant extract-based synthesis provides a simple pathway for nanoparticle formation, microbial synthesis—especially using photosynthetic microorganisms such as cyanobacteria and microalgae—has emerged as a more cost-effective and scalable alternative [176,177]. These photosynthetic microorganisms are ideal biofactories for nanoparticle production due to their high adaptability, low cultivation costs, and additional capacity for bioremediation [178]. Moreover, cyanobacteria and microalgae naturally produce a wide array of bioactive compounds—including phytohormones, vitamins, polysaccharides, and amino acids—which can promote seed germination, enhance root and vegetative development, improve nutrient uptake efficiency, and increase plant resilience to biotic and abiotic stresses [178].

Our analysis of the scientific literature over the past five years reveals several noteworthy trends:Despite ongoing discussions since the late 1990s about the potential of nanotechnology in agriculture, actual adoption, and application remain limited in this sector. For example, a bibliometric review of nanotechnology-related research from 2009 to 2019 found that only 10% of approximately 92,000 publications focused on agriculture [179]. Our updated analysis of publications from 2020 to 2024 confirms the continuation of this trend, indicating a persistent gap between laboratory innovation and agricultural implementation;The majority of literature examples related to the use of living organisms in nanosynthesis for agriculture focus primarily on plant-based materials;Although numerous nanoparticles have been successfully synthesized using cyanobacterial and microalgal strains, the vast majority of research has concentrated on medical and environmental applications rather than agriculture.

Actionable steps and future directions.

To fully harness the potential of bionanotechnology in sustainable agriculture, several targeted directions for research and development are recommended to advance the practical application of nanomaterials derived from cyanobacteria- and microalgae-mediated synthesis in sustainable agriculture:-Future efforts should concentrate on identifying and characterizing novel strains with enhanced metabolic profiles and high nanoparticle synthesis capacity, along with optimizing their genetic or metabolic factors to improve yield and reproducibility;-While numerous studies have demonstrated the feasibility of both in vivo and in vitro nanoparticle synthesis, comparative investigations conducted under controlled experimental conditions are still necessary to determine the most efficient, reproducible, and economically viable approaches for specific applications;-Equally important is the development of standardized protocols for nanoparticle characterization, stability assessment, and biological safety validation, aligned with international regulatory frameworks, to support future commercial applications;-Applied research should extend beyond laboratory or controlled environments by implementing semi-industrial studies to validate the effectiveness of biogenic nanoparticles in real agricultural systems. Investigations are needed into their interactions with soil microbiota and potential ecological effects to ensure safe and sustainable use;-Integrating these nanoparticles into smart delivery systems, such as nano-encapsulated biofertilizers or targeted biopesticide formulations, could significantly enhance application precision and reduce environmental impact;-The integration of such biotechnological processes into circular economy models may also represent a strategic future direction, provided that their efficiency and safety are validated at the application scale.

By advancing these strategies, bionanotechnology can become a transformative tool in the transition toward ecologically responsible and resource-efficient agriculture. With the dual advantages of enhanced productivity and reduced environmental impact, the integration of photosynthetic microorganisms in nanoparticle biosynthesis offers a compelling alternative to conventional agrochemicals. However, continued interdisciplinary research, regulatory oversight, and responsible application are essential to ensure the long-term sustainability and safety of these emerging technologies.

## Figures and Tables

**Figure 1 nanomaterials-15-00990-f001:**
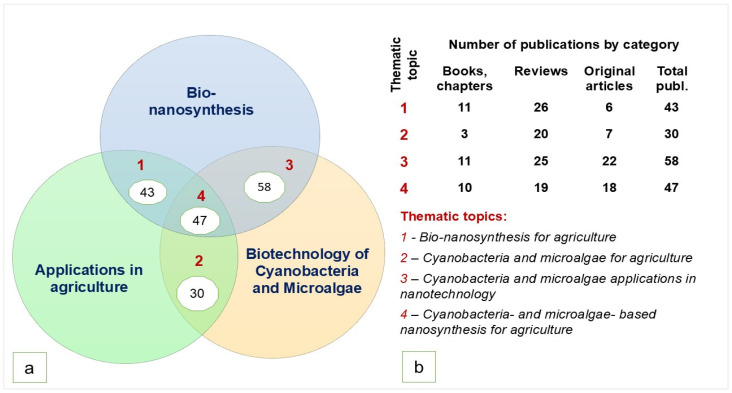
(**a**) Venn diagram grouping the eligible topics of the study. (**b**) Distribution of selected publications into thematic groups.

**Figure 2 nanomaterials-15-00990-f002:**
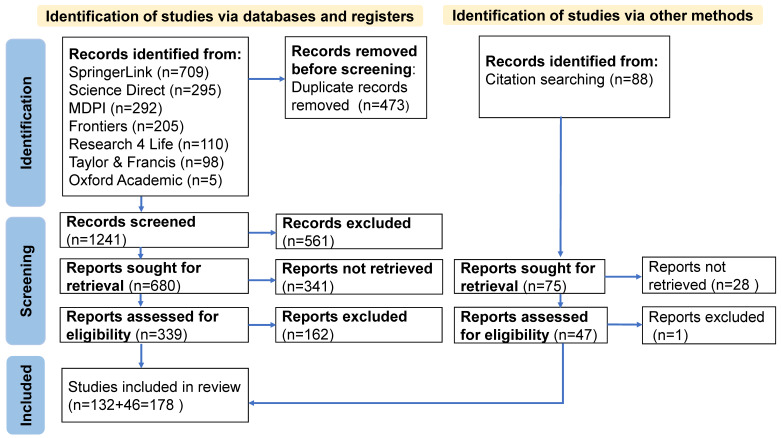
PRISMA flow diagram for data collection.

**Figure 3 nanomaterials-15-00990-f003:**
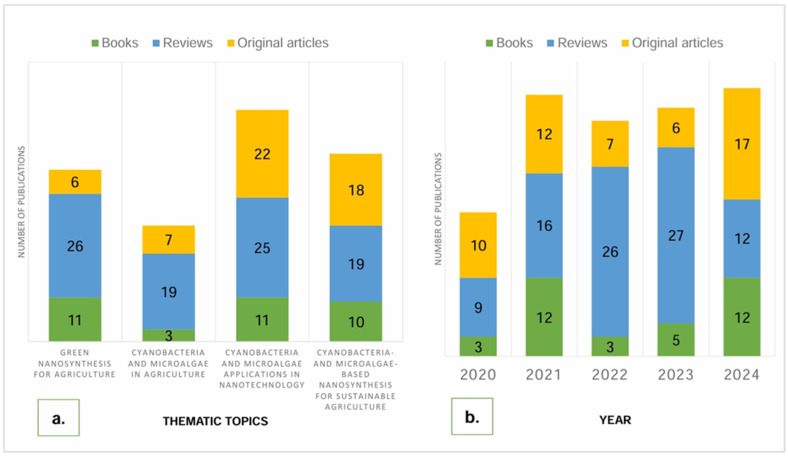
Distribution of selected publications by thematic topic (**a**) and by year (**b**).

**Figure 4 nanomaterials-15-00990-f004:**
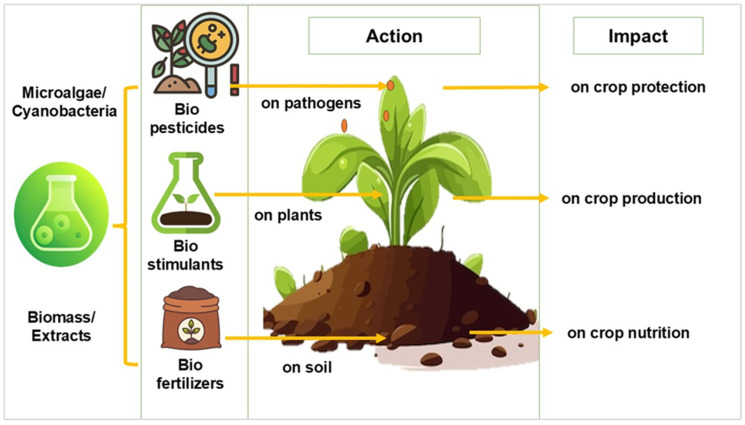
Mode of action of cyanobacteria- and microalgae-based products in agriculture. Cyanobacterial and microalgal biomass/extracts act as (i) biopesticides, suppressing plant pathogens to enhance crop protection; (ii) biostimulants, enhancing plant vigor to increase crop production; and (iii) biofertilizers, improving soil characteristics to enhance crop nutrition.

**Figure 5 nanomaterials-15-00990-f005:**
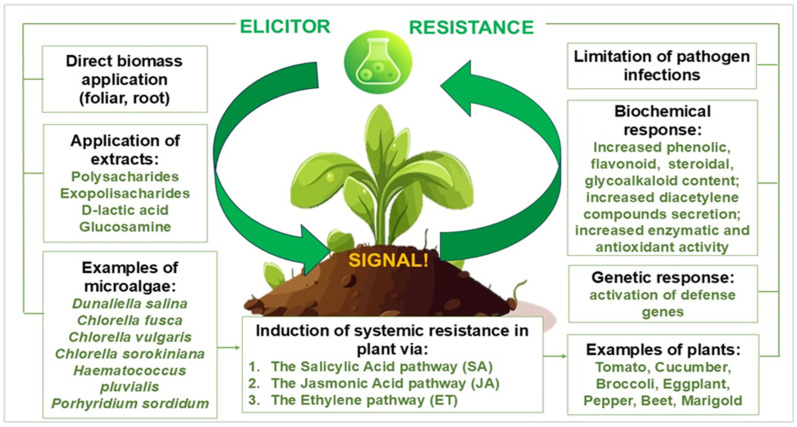
Microalgal elicitors in plant defense mechanisms. Microalgal biomass or extracts (e.g., polysaccharides, D-lactic acid, glucosamine) act as elicitors that trigger plant immune responses via the salicylic acid (SA), jasmonic acid (JA), and ethylene (ET) pathways. This leads to systemic resistance, activation of defense genes, and enhanced biochemical responses—including increased levels of phenolics, flavonoids, and antioxidants. As a result, foliar or root application limits pathogen infections and improves plant resistance.

**Figure 6 nanomaterials-15-00990-f006:**
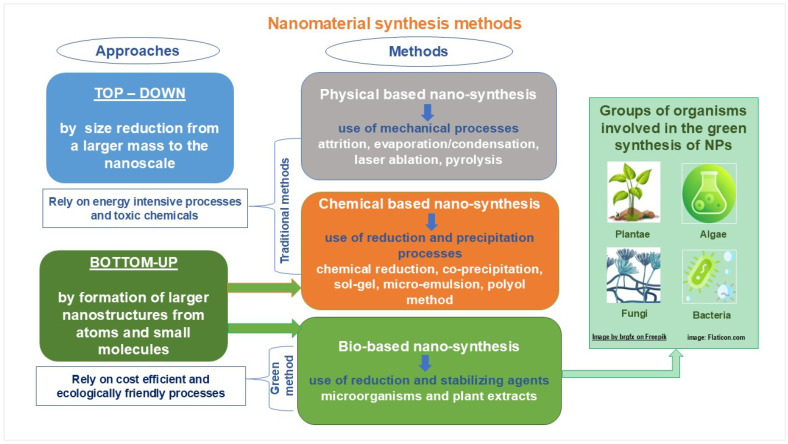
Methods employed in nanosynthesis. Nanosynthesis techniques categorized into two main approaches: (i) top–down methods involve breaking down bulk materials into nanoscale structures through physical processes, and (ii) bottom–up methods build nanoparticles from atomic/molecular units via chemical or biological processes. Traditional physical- or chemical-based nanosynthesis techniques are often energy-intensive and may involve toxic reagents, whereas green synthesis offers eco-friendly and cost-effective alternatives using organisms such as plants, algae, fungi, and bacteria.

**Figure 7 nanomaterials-15-00990-f007:**
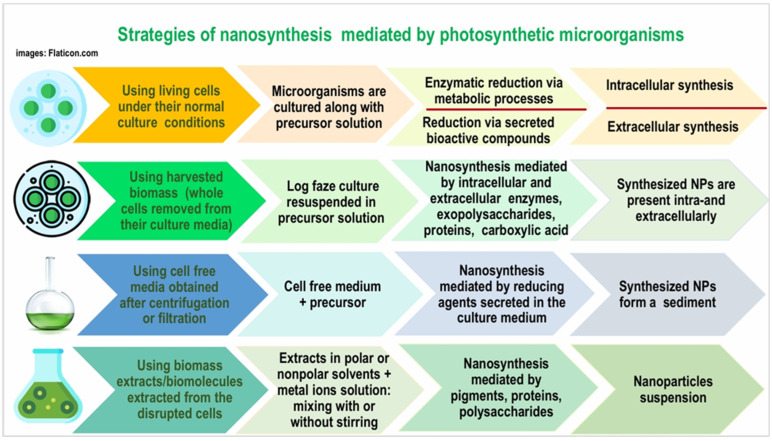
Techniques used for the synthesis of metallic NPs by cyanobacteria and microalgae. The figure illustrates four main biosynthesis approaches using photosynthetic microorganisms: (1) Live cells under normal culture conditions reduce metal ions via metabolic activity. (2) Whole-cell mediated synthesis, using harvested biomass to mediate nanoparticle (NP) formation through enzymatic and biomolecular interactions. (3) Cell-free supernatant synthesis, employing secreted biomolecules in the culture medium for extracellular NP synthesis. (4) Biomolecule-based synthesis, where cell extracts containing active compounds (e.g., proteins, pigments, polysaccharides) act as reducing and stabilizing agents.

**Figure 8 nanomaterials-15-00990-f008:**
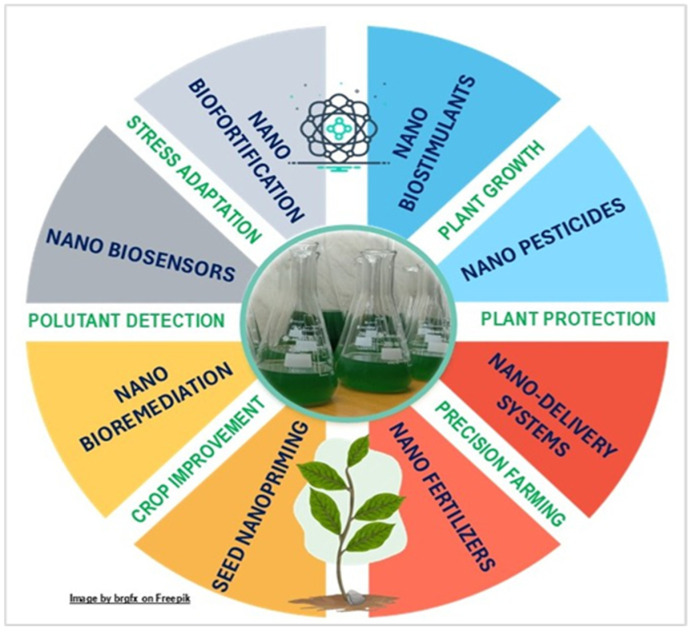
Phyco-bionanotechnological agro-solutions.

**Table 1 nanomaterials-15-00990-t001:** Key topics explored in publications on biogenic nanoparticles derived from cyanobacteria and microalgae.

Topic	Main Ideas
Bio-based mechanism and techniques of NPs synthesis	- Intracellular and extracellular biochemical reduction of metallic ions, supported by metabolic processes (nitrogen fixation, photosynthesis, cellular respiration); - Nanosynthesis can be performed using dried biomass or living cells - Techniques: (1) using algal or cyanobacteria suspension, (2) using cell filtrate, (3) using metabolites in the culture medium, and (4) using extracted biomolecules; - Stages of NPs synthesis: nucleation, development, and stabilization
Capping, reducing, and stabilizing agents	Carboxylic acid; polyphosphates; carbohydrates, including polysaccharides; proteins, including Histone H4; enzymes (NADH reductases, nitrate reductase, nitrite reductase); polyunsaturated fatty acids, antioxidants (polyphenols and tocopherols); pigments like carotenoids, chlorophylls, and phycobilins (phycocyanin, allophycocyanin and phycoerythrin)
Genera used for nanosynthesis	Cyanobacteria: *Anabena*, *Arthrospira/Spirulina*, *Aphanizomenon*, *Aphanocapsa*, *Aphanothece*, *Aulosira*, *Calothrix*, *Cyanothece*, *Cylindrospermopsis*, *Cylindrospermum*, *Desertifilum*, *Desmonostoc*, *Gloeocapsa*, *Leptolingbya*, *Limnotrix*, *Lyngbya*, *Microchaete*, *Microcoleus*, *Microcystis*, *Nannochloropsis*, *Nostoc*, *Oscilatoria/Phormidium*, *Plectonema*, *Porphyridium*, *Scytonema*, *Synechocystis*, *Synechococcus*, *Trichodesmium*, *Westiellopsis;* Microalgae: *Acutodesmu/Amphora*, *Asterarcys*, *Botryococcus*, *Chlamydomonas*, *Chlorella*, *Chlorococcum*, *Chromochloris*, *Coelastrum*, *Cosmarium*, *Desmochloris*, *Dunaliella*, *Euglena*, *Galdieria*, *Kirchneriella*, *Klebsormidium*, *Nannochloropsis*, *Neochloris*, *Pediastrum*, *Picochorum*, *Pithophora*, *Planophila*, *Rhizoclonium*, *Tetradesmus*, *Tetraselmis*, *Scenedesmus*, *Diatoms: Amphora*, *Chaetoceros*, *Diadesmus*, *Navicula*, *Nitzschia*, *Phaeodactylum*, *Skeletonema*, *Thalassosira*
Factors influencing the NPs synthesis	Cyanobacteria and microalgae species, precursor concentration, incubation period, illumination, pH, temperature, biomass concentration
Precursors (bulk material)	AgNO_3_, AgC_2_H_3_O_2_, AgClO_4_, Ag_2_SO_4,_ Au (S_2_O_3_)_2_, AuCl_4_, HAuCl_4_, Pd (NO_3_)_2_, H_2_PtCl_6_, Pd (NO_3_)_2_, CdCl_2_ and Na_2_S, Zn (NO_3_)_2_, ZnSO_4_
Precursor concentration	0.1–5 mM AgNO_3_, 0.5 mM HAuCl_4_, 0.1M FeCl_3_/FeCl_2_, 0.05M FeCl_3_
Bio reductant/precursor ratio, *v*:*v*	1:4, 1:2, 1:9, 9:1—for AgNO_3_
Incubation period	Minimal: 5 min; maximal: 1 month
pH	The diapason of 4–14 was applied, optimum 7, 4; 9; exceptionally 14
Temperature	Usually: 20–25 °C; exceptionally: 37 °C, 100 °C, 180 °C, 200 °C
Illumination	Synthesis was performed in both light and dark conditions, but the most successful synthesis was registered under illumination
Type of synthesized NPs	Metallic (Ag, Au, Pt, Cd, Pd), metalloid (Se); metallic oxide (CuO, ZnO, TiO_2,_ Fe_3_O_4_); metallic chalcogenides (CdS, HgS, ZnS) and other NPs (carbon nanodots, QDs, and Ag–Au nanoalloy, FeOOH, AgCl, Fe_3_O_4_/Ag nanocomposite)
Characteristics of nanoparticles	Size: from 2 nm to 200 nm Shape: spherical, elongated, triangular, cubic, pentagonal, hexagonal, octahedral, stars, nanorods
Possible application of synthesized NPs	Biomedical (drug delivery, anti-cancer, anti-bacterial, anti-fungal, anti-inflammatory, anti-hemolytic, anti-aging), environmental bioremediation, agricultural (nano-fertilizers), dye decolorization agent

**Table 2 nanomaterials-15-00990-t002:** Intracellular and extracellular synthesis of nanoparticles.

Feature	Intracellular Synthesis	Extracellular Synthesis
Location	Inside the cell (cytoplasm, cell membrane, thylakoid membrane)	Outside the cell (in the culture medium)
Process	Metal ions enter cells and are reduced internally	Biomolecules secreted by cells reduce metal ions in the surrounding medium
Mechanism	Cells absorb metal ions, which are then bio-reduced into NPs via metabolic processes	The secreted biomolecules act as reducing, capping, and stabilizing agents, promoting nanoparticle formation
Recovery	Requires cell lysis and extraction	Nanoparticles are already in the medium
Advantages	Controlled NP size and shape	Easier recovery

**Table 3 nanomaterials-15-00990-t003:** Nanomaterials for agriculture.

Application	Nanomaterials	References
Seed priming, growth, and reproduction stimulation	AgNPs, AlNPs, CeNPs, CuNPs, CuONPs, FeNPs, Fe_2_O_3_NPS, FeNPs with SiO_2_, MgONPs, multi-walled carbon nanotubes (MWCNTs), NiONPs, quantum dots (QD), SeO_2_NPs, TiO_2_NPs, ZnONPs	[98,99,100]
Fertilizers	AlNPs, B_2_O_3_NPs, nano-CaCO_3_, carbon nanotubes (CNT), CuNPs, CuONPs, Cu-Chitosan NPs, copper nanowires, FeNPs, FeONPs, FeO(OH)NPs, Fe_2_O_3_NPs, Fe_3_O_4_ NPs, MgNPs, MgONPs, MnNPs, MoNPs, MWCNTs, KFeO_2_NPs, SeNPs, SiNPs, SiO_2_NPs, TiO_2_NPs, ZnNPs, ZnONPs, Zn-chitosan NPs, ZnNPs, hydroxyapatite NPs, 2-D graphite carbon NPs, vermiculite, nanoclay, zeolite	[1,99,101,102,103,104,105,106,107,108]
Fungicides	AgNPs, alumino silicates (3Al_2_O_3_·2SiO_2_) NPs, CNTs, CuNPs, FeNPs, MgNPs, MgONPs, NiNPs, SeNPs, SiNPs, Silver-coated carbon nanotubes hybrid NPs, TiO_2_NPs, ZnONPs	[99,106,109,110]
Bactericides	AgNPs, Ag, Cu and Ti nanocomposites, Al_2_O_3_NPs, CuNPs CuONPs, graphene oxide, Fe_3_O_4_–Ag core–shell magnetic NPs, MgONPs, TiO_2_NPs, ZnONPs	[99,106,110]
Herbicide nano-carriers	Carboxymethyl chitosan-modified carbon NPs, chitosan NMs, poly ε-caprolactone, nanocapsules	[99]
Insecticides	AgNPs, CuNPs, CuONPs, NiONPs, PbNPs, SiO_2_NPs, ZnONPs, TiO_2_NPs, electrospun nanofibers	[99,111]
Hit stress tolerance	AgNPs, CaNPs, CeONPs, SeNPs, SiNPs, TiO_2_ NPs, ZnONPs	[99,108]
Drought stress tolerance	AgNPs, CuNPs, CNT, Fe_3_O_4_ NPs, fullerenes, ZnNPs, MWNTs, SiNPs, single-walled carbon nanotubes (SWNTs), TiO_2_NPs, ZnONPs,	[96,99,106,108]
Salt stress tolerance	AgNPs, CeONPs, FeNPs, Fe_3_O_4_NPs, MWCNT, SiNPs, TiO_2_NPs	[99,108]
Metal stress tolerance	FeONPs, SiO_2_NPs, ZnONPs	[99,108]
Gene delivery	Cationic polymers, CuNPs, dimethylamino ethyl methacrylate (DMAEM) polymer NPs, lipid NPs (LNPs), nanotubes, quantum dots (QDs)	[99]
Nanosensors	AgNPs, AuNPs, CdTeQD4-Rd, CNT, CuNPs, MWCNTs, SWNTs, fullerenes, graphene oxide NPs, nanoTiO_2_/nafion composite, nanoscale wires, Pt nanoparticle-anchored zirconium-based metal–organic framework nanocomposites, SiO_2_NPs, silicate/glucose oxidase, TiNPs, ZnONPs, ZnONP–chitosan nanocomposite QDs	[99,102,106,107,112,113,114]
Vegetable and fruit preservation	AgNPs, cellulose nanocrystal, chitosan-assisted nano-silica, chitosan film-based nano-SiO_2_, nano-ZnO	[103,109]

**Table 4 nanomaterials-15-00990-t004:** Bionanotechnological aspects of cyanobacteria and microalgae applications in agriculture.

Cyanobacterial/ Microalgal Strain	NM Name and Characteristics	Synthesis Methods	Crops	Outcomes	Ref.
Application of nanoparticles synthesized by photosynthetic microorganisms in the agri-food sector					
*Chlorella* K01	Fe_3_O_4_ NPs, 76.5 nm	Using microalgal aqueous extract, t = 65 °C, pH = 6–12	- Rice, maize, mustard, green gram, watermelon seed priming	- Stimulatory effect on germination and seedling vigor index ranging from 35% to 100% above control, maximal effect observed in green gram - Antifungal activity against *Fusarium, Rhizoctonia,* and *Pythium*	[67]
*Leptolyngbya* sp. WUC 59	Spherical Ag NPs, 20–35 nm	Using cell-free aqueous extract, t = 70 °C, pH = 6–12	*Triticum aestivum* L. seed priming	- Increase in the root and shoot length of 3.0 cm and 8.0 cm at 25 mg L^−1^ concentration of Ag NPs and decrease at higher concentrations	[72]
*Chlorella vulgaris*	Spherical Ag NPs, 5.76 nm	Using solution of soluble polysaccharides, 24 h, in the dark	*Triticum vulgare* *Phaseolus vulgaris* seed priming	- Increase in shoot height by apr. 23%, root length by apr. 30%, and first vegetative leaves by apr. 60% above control level	[134]
*Desmonostoc alborizicum*	Spherical Se NPs, 58.8 nm	Using cell-free extract, t = 60 °C, mechanical stirring	-	- Antifungal activity against plant pathogens. *Alternaria alternata* showed the highest sensitivity	[92]
*Nodosilinea nodulosa*	Spherical Co_3_O_4_ NPs, 41 nm	Using cell-free extract, t = 60 °C, mechanical stirring	-	- Antifungal activity against *Fusarium oxysporum* with a zone of inhibition of 3 ± 0.04 µg/mL, *Aspergillus flavus* with a zone of inhibition of 2 ± 0.04 µg/mL	[93]
*Nostoc* sp.	Spherical ZnO NPs, 18.47 nm	Using acetone extract, T = 60 °C, mechanical stirring 1 h	corn, wheat seed priming	- Enhanced corn seedlings parameters by apr. 60–75% and chlorophyll content by 30% at NP concentr. of 10 µg/mL; wheat seedling parameters by 140–170% and chlorophyll content by 60% at NP concentr. of 15 µg/mL	[135]
Nanoparticle-enriched extracts of cyanobacteria and microalgae in plant biotechnology					
*Calothrix elenkinii*	AgNPs	NPs from Sigma-Aldrich Chemical Pvt. Ltd. applied in synergy with cyanobacterium	*Lycopersicon esculentum* (tomato) detached leaf assay	Biocontrol efficacy against *Alternaria alternata*: higher leaf chlorophyll accumulation and lower endoglucanase activity	[136]
*Calothrix elenkinii*	AgNPs	NPs from Sigma-Aldrich Chemical Pvt. Ltd. applied in synergy with cyanobacterium	*Lycopersicon esculentum* Mill(tomato) foliar application on plants infected by *Alternaria alternata*	- Disease severity reduced by 47–58% - Leaf chlorophyll, carotenoid content, polyphenol oxidase activity significantly increased by 44–45%, 40–46%, and 23–33%, respectively - Ergosterol content decreased by 63–79%	[137]
*Spirulina* sp.	CuNPs	Chemically synthesized NPs, combined with spirulina extract	French basil foliar treatment 2 seasons, pot experiment in natural field conditions	- Maximum effects were achieved with 500 mg/L CuNPs and 1.5 g/L spirulina: a 4–5-fold increase in oil yield, the highest chlorophyll and carotenoid levels, and a 45% increase in plant height compared to the control	[138]
*Arthrospira platensis*	CuNPs CuONPs 100 nm	Cyanobacterium cultivated with NPs (Merck KGaA)	Triticale seed priming	- Extracts from cyanobacterium cultured in media supplemented with CuONPs increased chlorophyll content in leaves by 42.8% and carotenoid content by 37.4% - Extracts from biomass cultured with CuNPs increased antioxidant activity in triticale leaves by 48–65.3%	[139]
*Cyanothece* sp.	Graphene, graphene oxide, carbon nanotubes	Chemically synthesized NMs, in combination with methyl salicylate and *Cyanothece* sp. and *Enterobacter cloacae*	*Hordeum vulgare*, *Vicia faba* adding to soil, presoaking in biofertilizer	- The germination of both plants in the presence of graphene and carbon nanotubes, combined with biofertilizer treatment under the highest salinity stress level, almost completely alleviated the effects of salinity stress,	[140]
*Scenedesmus obliquus*	ZnONPs	Biosynthetized ZnONPs, using cell-free algal extract incubated overnight at 28 °C, applied along with microalga extract	Tomato, foliar spray	Algal extract + bio-ZnONPs increased total fruit yield by 37% and marketable yield by 43.1%; raised soil microbial and dehydrogenase activity by 369.4% and 298.8%, respectively; boosted zinc accumulation by 74.8% in roots, 182.7% in leaves, and 104.2% in fruit	[141]
*Spirulina* sp.	Spindle-shaped FeNPs, 24.77 ± 5.264 nm	Biosynthetized FeNPs (using spirulina biomass), dried spirulina biomass, nano-iron-loaded spirulina biomass	Rice, fertilizer type	Nano-iron-loaded *Spirulina* sp. biomass increased shoot length by 37.6%, crop productivity by 47%, and grain weight by 15%	[58]
*Chlorella* sp. MF1	AgNPs	Using algal biomass aqueous extract 28 °C, 1 h, pH-6	*Eruca sativa* seed soaking, foliar spray	- Increase in root length, fresh weight, and dry weight in response to increased concentrations of AgNPs and Chlorella	[142]
*Arthrospira* *platensis*	Lecithin/chitosan NPs	Nanoliposome system employing lecithin/chitosan as carriers of *Arthrospira* protein hydrolysate	Tomato, weekly sprays	NPs and protein hydrolysate treatments significantly stimulated plant growth, up to a 49.5% increase in plant height during the vegetative phase	[143]
Cyanobacteria- and microalgae-based nanosensors					
*Chlamydomonas reinhardtii* UV180	Nano-formulated atrazine, encapsulated into zein and chitosan poly-ε-caprolactone nanoparticles	Whole algal cells immobilized on carbonized lignin electrodes and integrated into a photo-electrochemical transducer	-	The biosensor demonstrated a linear response for atrazine detection in the 0.1–5 μM range, with detection limits of 0.9 nM and 1.1 nM for atrazine–zein and atrazine-PCL-Ch nanoparticles, respectively	[144]

## Data Availability

Not applicable.

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
