# Peer review of "Photosynthetic Microorganisms and Biogenic Synthesis of Nanomaterials for Sustainable Agriculture"

_nanomaterials, 2025, doi:10.3390/nano15130990_

Round 1
Reviewer 1 Report
Comments and Suggestions for Authors
It presents a systematic review on an interesting topic with broad application potential. The scientific literature was reviewed between 2020 and 2024, and the main results of several of the cited references are summarized. However, it does not strike me as an innovative review that goes beyond what is proposed in published articles; probably because the subject matter is primarily methodological and technical.
On line 42: Is it necessary to expand on the biochemical and molecular mechanisms of adaptation and tolerance exhibited by these microorganisms, which make them so attractive for various applications?
On lines 267 and 663... the hormonal group or the word that should be written is Auxins, not the name of the indole-3-acetic acid molecule.
Reviewer 2 Report
Comments and Suggestions for Authors
- The review should primarily center on "cyanobacteria and microalgae, as well as their biologically synthesized nanomaterials, for agricultural applications."The discussion must remain tightly focused on this theme, avoiding excessive elaboration on general nanotechnology.
Section 3.1 can be removed or integrated into other relevant sections.
Sections 3.2–3.4 should be condensed to maintain conciseness.
- Additional Revisions Required:
Line 28: "10⁻⁹ m" should be converted to nanometers (nm) for clarity.
Page 3 (Figure 1): The data in the figure lacks units—please ensure proper labeling.
The data for Region 1 in Figure 1b does not match Region 1 in Figure 1a—consistency must be verified.
Line 84: "Eligible organisms" should be formatted consistently (e.g., bolded) to match preceding text.
Line 860: The subheading format is inconsistent with previous sections—please standardize.
Page 26 (Conclusions): The conclusion should highlight the future prospects of cyanobacteria and microalgae in agricultural nanotechnology. Actionable recommendations for further research and development should be included.
Reviewer 3 Report
Comments and Suggestions for Authors
This manuscript offers a comprehensive and well-organized review of the role of photosynthetic microorganisms (cyanobacteria and microalgae) in the green synthesis of nanomaterials and their applications in sustainable agriculture. It integrates recent findings from the past five years and follows PRISMA guidelines for systematic reviews, which adds to its scientific rigor.
The review is thorough, with extensive coverage of the topics, supported by numerous references, and presents a useful classification of nanomaterials and synthesis methods. It has the potential to serve as a valuable reference for researchers in nanobiotechnology and agricultural sciences. However, to enhance the scientific clarity, accessibility, and impact of the manuscript, I suggest the following improvements:
1) Some sections (e.g., Introduction and Section 3.1) contain redundant sentences or concepts repeated in different wording. For example, the ability of cyanobacteria and microalgae to synthesize nanomaterials is mentioned multiple times with little added value.
2) It is better to include a table summarizing key quantitative outcomes (e.g., typical nanoparticle sizes, crop yield improvements, or stress tolerance changes with NPs).
3) It is better to give a section about the author's own opinions based on the future research and directions.
4) the caption of figures should be much more informative, such as Figures 5-7, there are many information in figure but hard to read from captions.
5) The review is focused on the effect of nanomaterials on agriculture, however, there is no any table or figure giving the direct and quantitative values of real-world agricultural outcomes, such as 25% higher or something.
Round 2
Reviewer 3 Report
Comments and Suggestions for Authors
I feel all the comments have been addressed well, and the current version could be accepted without further modification.